# β-Catenin/TCF4 Complex-Mediated Induction of the *NRF3* (*NFE2L3*) Gene in Cancer Cells

**DOI:** 10.3390/ijms20133344

**Published:** 2019-07-08

**Authors:** Shiori Aono, Ayari Hatanaka, Atsushi Hatanaka, Yue Gao, Yoshitaka Hippo, Makoto Mark Taketo, Tsuyoshi Waku, Akira Kobayashi

**Affiliations:** 1Laboratory for Genetic Code, Graduate School of Life and Medical Sciences, Doshisha University, Kyotanabe, Kyoto 610-0394, Japan; 2Japan Society for the Promotion of Science, Tokyo 102-0083, Japan; 3Division of Molecular Carcinogenesis, Chiba Cancer Center Research Institute, Chiba, Chiba 260-8717, Japan; 4Division of Experimental Therapeutics, Graduate School of Medicine, Kyoto University, Kyoto 606-8501, Japan; 5Laboratory for Genetic Code, Department of Life and Medical Sciences, Faculty of Life and Medical Sciences, Doshisha University, Kyotanabe, Kyoto 610-0394, Japan; 6Laboratory for Genetic Code, Graduate School of Life and Medical Sciences, and Department of Life and Medical Sciences, Faculty of Life and Medical Sciences, Doshisha University, Kyotanabe, Kyoto 610-0394, Japan

**Keywords:** NRF3, NRF2, β-catenin, Wnt signaling, GLUT1, transcription, colorectal cancer, organoid

## Abstract

Remarkable upregulation of the NRF2 (NFE2L2)-related transcription factor NRF3 (NFE2L3) in several cancer tissues and its correlation with poor prognosis strongly suggest the physiological function of NRF3 in tumors. Indeed, we had recently uncovered the function of NRF3, which promotes cancer cell proliferation by p53 degradation via the 20S proteasome. Nevertheless, the molecular mechanism underlying the induction of *NRF3* gene expression in cancer cells is highly elusive. We herein describe that *NRF3* upregulation is induced by the β-catenin/TCF4 complex in colon cancer cells. We first confirmed high *NRF3* mRNA expression in human colon cancer specimens. The genome database indicated that the human *NRF3* gene possesses a species-conserved WRE sequence (TCF/LEF consensus element), implying that the β-catenin/TCF complex activates *NRF3* expression in colon cancer. Consistently, we observed that the β-catenin/TCF4 complex mediates *NRF3* expression by binding directly to the WRE site. Furthermore, inducing NRF3 activates cell proliferation and the expression of the glucose transporter *GLUT1*. The existence of the β-catenin/TCF4-NRF3 axis was also validated in the intestine and organoids of *Apc*-deficient mice. Finally, the positive correlation between *NRF3* and β-catenin target gene expression strongly supports our conclusion. Our findings clearly demonstrate that NRF3 induction in cancer cells is controlled by the Wnt/β-catenin pathway.

## 1. Introduction

The development of human cancer genome databases such as The Cancer Genome Atlas (TCGA) and Oncomine have led directly to the discovery of significant functional mutations in genes and pathways [1,2]. This information also indicates that the biological roles and regulatory mechanisms of numerous genes in tumorigenesis are not defined. Thus, determining these unknown roles will be indispensable for the development of promising targeted molecules for new cancer therapies and precision medicine.

In this regard, we focused on the transcription factor NRF3 (NFE2L3), because it is highly expressed in several cancers, including colorectal adenocarcinoma, and is correlated with poor prognosis [3]. NRF3, which was originally discovered by our group [4], belongs to the cap “N” collar (CNC) family of six transcription factors, including NRF2 (NFE2L2) and NRF1 (NFE2L1); NRF2 is a famous cancer driver gene and an oxidative stress response transcription factor, and NRF1 sustains proteostasis by mediating proteasome gene expression [5,6,7]. The physiological function of NRF3 remains unknown; *Nrf3* knockout mice do not show obvious abnormalities [8,9,10]. Recently, we found that NRF3 promotes the proliferation of cancer cells by degrading the tumor suppressor p53 through 20S proteasome activation [11] (manuscript submitted by Waku et al.). In support of our findings, accumulating evidence indicates that NRF3 regulates the growth and metastasis of thyroid, pancreatic, testis and breast cancer [12,13,14,15,16]. Nevertheless, the molecular mechanism underlying *NRF3* gene induction in cancer cells is poorly understood.

A large body of evidence has revealed that the accumulation of multiple gene mutations underlies the development of cancer [17]. For instance, the Wnt/β-catenin pathway, which is essential for normal intestinal growth and development, plays a key role in the initiation of colorectal cancer progression [18,19,20]. Inactivation of the adenomatous polyposis coli (*APC*) gene and the *CTNNB1* gene encoding β-catenin is an initial event leading to the development of adenoma in most sporadic cases. These genetic alterations stabilize the β-catenin protein and promote its nuclear entry, leading to the activation of gene expression by T-cell factor/lymphoid enhancer-binding factor (TCF/LEF) family proteins binding to the WRE sequence (Wnt response element, TCF/LEF consensus element). The TCF/LEF family consists of TCF1, TCF3, TCF4 and LEF1. The upregulation of β-catenin/TCF target genes, such as *c-MYC*, *AXIN2* and *LGR5,* regulates cell proliferation and stemness in epithelial stem and progenitor cells [21,22,23]. Thus, constitutive activation of β-catenin by *APC* mutations gives rise to hyperplastic epithelium. Further carcinoma development requires an activating mutation in the *K-ras* gene and loss of the *p53* gene.

Gene alterations dramatically change the metabolic status of cancer cells [21,24,25]. One type of metabolic reprogramming in cancer cells is the Warburg effect, which is characterized by a shift toward glycolysis from oxidative phosphorylation. Indeed, certain cancer cells upregulate the gene expression of the glucose transporters GLUT1 (SLC2A1) and GLUT3 (SLC2A3) to increase glucose uptake, which correlates with poor prognosis.

We herein report that the molecular mechanisms of *NRF3* induction in cancer cells involve the β-catenin/TCF4 complex. We first identified a species-conserved WRE site that is recognized by the β-catenin/TCF4 complex in the *NRF3* gene. The β-catenin/TCF4 complex mediates *NRF3* expression in several cancer cells through binding to its WRE site. We also found that upregulating NRF3 augments cell proliferation and the expression of the *GLUT1* gene. The biological significance of the β-catenin/Tcf4-Nrf3 axis was confirmed by deletion of the *Apc* gene in the mouse intestine and its derived organoids. Finally, we found a positive correlation between *NRF3* and *c-MYC* or *LGR5*, target genes of the β-catenin/TCF4 complex. Altogether, this study reveals the β-catenin/TCF4-NRF3 axis as the molecular basis for *NRF3* gene upregulation in cancer cells.

## 2. Results

### 2.1. High Expression of The NRF3 Gene in Human Colorectal Cancer Specimens

To further examine the biological relevance of NRF3 in cancer, we first conducted an NRF3 expression analysis using a human cancer RNA dot blot (Figure 1). The blot possesses RNA derived from both tumor (T) and adjacent normal (N) tissues for various human cancers. Intriguingly, *NRF3* mRNA expression was induced in several types of cancer, especially in colon and rectal carcinoma. These results strengthen our hypothesis that NRF3 is involved in colorectal carcinogenesis. 

### 2.2. The β-catenin/TCF4 Complex Directly Mediates NRF3 Gene Induction in Cancer Cells

To investigate the molecular mechanism underlying *NRF3* gene induction in colorectal cancer cells, we considered possible regulation by the β-catenin/TCF complex. Mutations in the Wnt-β-catenin pathway are crucial initial events in colon carcinogenesis [18,19,20]. We searched the regulatory region of the *NRF3* gene and successfully found a species-conserved WRE sequence as the TCF recognition element in its first intron (Figure 2A).

To further confirm our hypothesis, we explored whether β-catenin/TCF binds directly to the WRE sequence in the *NRF3* gene. In the TCF/LEF family, TCF4 was reported to be abundantly expressed in colon cancer cells [19,26]. Supporting this notion, we observed that *TCF4* mRNA levels were much higher than *LEF1* mRNA levels in HCT116 cells (Figure 2B), implying that TCF4 could be a regulator of *NRF3* gene expression. We next carried out a chromatin immunoprecipitation analysis to assess the direct binding of both β-catenin and TCF4 to the WRE sequence (Figure 2C). As expected, the data clearly demonstrated the recruitment of β-catenin/TCF4 to the *NRF3* gene.

We addressed the question of whether the β-catenin/TCF4 complex mediates *NRF3* expression (Figure 2D–G). For this purpose, we used two different approaches. First, we examined the effects of *β-catenin* or *TCF4* knockdown on NRF3 expression in HCT116 cells. siRNAs against these two genes significantly reduced *NRF3* expression at both the mRNA and protein levels. Next, we chemically inhibited the β-catenin/TCF4 complex with iCRT14, which attenuates its DNA binding activity [27] (Figure 2H). As expected, iCRT14 treatment substantially reduced *NRF3* mRNA levels. Collectively, these results strongly indicate that the β-catenin/TCF4 complex is a direct regulator of *NRF3* gene induction in colon cancer cells.

### 2.3. The β-catenin/TCF4-NRF3 Axis Is Conserved in Other Cancer Cells

We also investigated β-catenin/TCF4-mediated *NRF3* regulation in other human cancer cell lines: DLD-1 (colorectal adenocarcinoma), H1299 (non-small cell lung carcinoma), A172 (glioblastoma) and HeLa (cervical epithelioid carcinoma) (Figure 3A). All these cell lines, except for HeLa cells, harbor mutations in the Wnt-β-catenin system, resulting in constitutive β-catenin activation [28]. As such, both *β-catenin* and *TCF4* siRNA suppressed *NRF3* mRNA expression in A172 and H1299 cells but not in HeLa cells. In DLD-1 cells, *β-catenin* siRNA but not *TCF4* siRNA caused a reduction in *NRF3* expression, implying the possibility that β-catenin activates NRF3 expression through other TCF family members. Supporting these results, these cell lines, but not HeLa cells, showed nuclear β-catenin accumulation, implying that it was transcriptionally active (Figure 3B and Appendix A). Moreover, we addressed the biological relevance of the β-catenin/TCF4 complex in relation to the NRF3 homologue NRF1 (Figure 3C). The data revealed that *β-catenin* or *TCF4* knockdown did not affect the expression of *NRF1*. These data indicate that the β-catenin/TCF4-*NRF3* axis is present in other tissue-derived cancer cells that harbor mutations in the Wnt pathway.

### 2.4. NRF3 Activates Cell Proliferation and GLUT1 Gene Expression in Colon Cancer Cells

Our finding highlighted another question: what is the biological function of induced NRF3 in cancer cells? We previously revealed that *NRF3* knockdown reduced DLD-1 cell growth [11]. Similar experiments using HCT116 cells demonstrated that *NRF3* siRNA significantly reduced cell growth (Figure 4A). To identify direct NRF3 target genes in cancer cells, we searched ChIP sequence data for MAFF and MAFK in the ChIP-Atlas database [29] because ChIP sequence data for NRF3 were not available and because these sMaf proteins are heterodimerization partners of NRF3. Intriguingly, we discovered MAFF- and MAFK-ChIP peaks on three species-conserved AREs in the *GLUT1* (*SLC2A1*) gene (Figure 4B). Because GLUT1 is well known to uptake glucose for the proliferation of glycolytic cancer cells [24,25], it is possible that NRF3 promotes cell proliferation by mediating *GLUT1* expression. To validate this possibility, we first performed ChIP analysis to determine the recruitment of NRF3 to these ARE sites (Figure 4C). The results clearly demonstrated that NRF3 binds to these ARE sites in the *GLUT1* gene enhancer. We next conducted siRNA knockdown experiments (Figure 4D). *NRF3* siRNA clearly reduced *GLUT1* expression at both the mRNA and protein levels. We had found that NRF3 promotes the degradation of the tumor suppressor p53 through activating the 20S proteasome [manuscript submitted by Waku et al.]. In addition, p53 is reported to downregulate *GLUT1* gene expression [30]. To examine the possibility that NRF3 activates *GLUT1* expression by downregulating p53, we conducted a double knockdown experiment and found negative results (Appendix A). We thus conclude that NRF3 leads to the upregulation of the *GLUT1* gene in cancer cells.

### 2.5. The β-catenin/Tcf4-Nrf3 Axis Is Conserved in Mouse Intestinal Tumors

To further validate our finding regarding the β-catenin/TCF4-NRF3 axis, we performed in vivo experiments using mouse intestine. *Apc*Δ716 mice develop intestinal tumors due to a null mutation in the *Apc* gene, thereby activating β-catenin [31]. Upon analyzing intestinal tumors in *Apc*Δ716 mice, we observed significant upregulation of *Nrf3* expression and the β-catenin target gene *c-Myc* (Figure 5A).

We next conducted similar experiments using mouse intestine-derived organoids to eliminate nonepithelial cell contamination. Organoids are considered to recapitulate physiological tissues better than a conventional two-dimensional (2D) culture system because of their self-organizing structures [32,33,34]. We generated mouse intestine-derived organoids lacking the *Apc* gene (KO) by infecting *Apc*^flox/flox^ intestine organoids (f/f) with *Cre*-expressing lentivirus (Figure 5B). Consistent with previous reports [35,36], *Apc* deletion induced the β-catenin target genes *c-Myc* and *Axin2* as well as the intestine epithelial stem cell markers *Lgr5* and *Cd44* (Figure 5C). Importantly, *Apc* KO organoids showed high induction of the *Nrf3* gene. We also observed significant upregulation of the *Glut1* gene as well as the *Gclm* and *xCt* genes, which are known to contain AREs, in *Apc* KO organoids. Alternatively, loss of *Apc* did not induce the expression of *Nrf1*, *Nrf2* or its target gene *Nqo1* (Figure 5D). Our findings suggest that the β-catenin/TCF4-NRF3 axis is conserved in mouse intestinal tumors.

### 2.6. Correlation of Gene Expression between NRF3 and Target Genes of The β-catenin/TCF4 Complex in a Human Cancer Database

Finally, we searched the GEPIA database to confirm the physiological relevance of the association between the Wnt-β-catenin pathway and NRF3 (Figure 6). Because *c-MYC* and *LGR5* are direct target genes of the β-catenin/TCF4 complex, the analysis showed a significant correlation among them. We thus conclude that *NRF3* is induced by the β-catenin/TCF4 complex in human cancer cells.

## 3. Discussion

Our present study identified that the β-catenin/TCF4 complex in the Wnt signaling pathway activates *NRF3* mRNA expression, consequently leading to the upregulation of cell proliferation and *GLUT1* gene expression in cancer cells (Figure 7). Tumors evolve by acquiring a series of sequential gene mutations that endow cells with the hallmarks of cancer, including resistance to apoptosis and self-sufficiency in growth signals [17]. The mutations that confer a selective growth advantage to the cell are called “driver” mutations. For instance, mutation of the tumor suppressor *APC* gene in the Wnt/β-catenin pathway is an initial and crucial event in normal epithelial cells that initiates human sporadic colorectal tumorigenesis [37]. Additionally, it has been reported that *APC* mutations cause global metabolic reprogramming of cells by inducing the expression of the protooncogene *c-MYC*, a β-catenin target gene [21,22,23]. Based on these results, we surmise that NRF3 plays a physiological role in the initiation of tumorigenesis.

We recently discovered that NRF3 activates the proliferation of cancer cells by promoting the degradation of the tumor suppressor p53 through activating the 20S proteasome [11] (Waku et al., under submission). Inhibition of the tumor suppressor function of p53 is required to acquire resistance to cell death, a crucial cancer hallmark. Nevertheless, it has been reported that *p53* gene mutations arise at the final step of colorectal tumor development, eventually generating malignant carcinoma (Figure 7) [17]. Thus, we also consider that one biological function of NRF3 might be the suppression of p53 function before its genetic inhibition at the final stage of cancer onset.

Sustaining the stemness of intestine epithelial stem and progenitor cells is indispensable for the development of colorectal tumors [18]. To this end, the β-catenin/TCF4 complex augments the expression of stemness-related genes, such as the *Axin2* and *Lgr5* genes, thereby promoting the proliferation of stem/progenitor cells. Consistently, we showed that *Apc* deletion in mouse *Apc*^flox/flox^ organoids substantially induced the expression of these genes along with the *Nrf3* gene (Figure 5C). It is also possible that NRF3 plays biological roles in sustaining the stemness of intestine epithelial stem/progenitor cells.

It appears that β-catenin/TCF4 merely mediates the expression of NRF3 target genes by inducing *NRF3* mRNA. However, NRF3 can be considered a stress-inducible transcription factor because of our following three observations: (1) Under physiological conditions, NRF3 is sequestered in the endoplasmic reticulum (ER), thereby preventing its nuclear translocation and transcriptional activity [38]. (2) The ERAD-related E3 ubiquitin ligase HRD1 promotes the cytoplasmic degradation of NRF3 [11]. (3) The nuclear translocation of NRF3 requires the proteolytic enzyme DDI2, through a similar mechanism as that of NRF1 [11] (arrow and arrowheads in Figure 2E,G and Figure 4D). This evidence allows us to hypothesize that certain unidentified stresses and/or signals trigger both cytoplasmic sequestration escape and nuclear entry of NRF3. This hypothetical mechanism may explain in part why *Nrf3* knockout mice do not show apparent abnormalities under physiological conditions [8,9]. Considering that *APC* mutations dramatically change the cell metabolic status by inducing the *c-MYC* gene, some metabolites or activated signal transduction pathways may activate the nuclear translocation of NRF3. For a comprehensive understanding of the physiological function of NRF3 in cancer cells, the identification of intrinsic NRF3 activation and/or stress signals remains a crucial issue.

We found that the glucose transporter *GLUT1* is a candidate NRF3 target gene in colon cancer cells. This finding is reasonable because cancer cells dramatically change their metabolic status from oxidative phosphorylation to glycolysis by activating *c-MYC*, a β-catenin target gene [21]. We need to explore whether NRF3-mediated *GLUT1* expression confers glucose uptake and proliferation abilities to cancer cells. Unfortunately, we could not show that *GLUT1* knockdown reduces glucose uptake by HCT116 cells (data not shown). Because *GLUT3* mRNA is also induced in an NRF3-independent mechanism in cancer cells [24,25], GLUT3 might compensate for the loss of GLUT1 function. Further examination is required to determine the effects of NRF3-mediated *GLUT1* gene expression on cancer cells.

## 4. Materials and Methods 

### 4.1. Reagents and Antibodies

iCRT14 was purchased from Sigma-Aldrich (SML0203) (St. Louis, MO, USA). The antibodies utilized were anti-β-catenin (610153; BD Biosciences, San Jose, CA, USA), anti-TCF4 (C48H11; Cell Signaling Technology, Danvers, MA, USA), anti-α-Tubulin (DM1A; Sigma-Aldrich, St. Louis, MO, USA), and anti-histone H3 (06-755; Millipore, Billerica, MA, USA). Normal mouse IgG (Santa Cruz Biotechnology, Dallas, TX, USA) and normal rabbit IgG (Wako Pure Chemicals, Osaka, Japan) were used for chromatin immunoprecipitation analysis. A monoclonal NRF3 antibody (#9408) raised against human NRF3 (amino acids 364-415) was generated as described previously [11].

### 4.2. RNA Dot Blot Analysis

The Human Multiple Tissue Expression (MTE^TM^) Array, which contains dots of polyA^+^ RNA normalized to ubiquitin from 76 different human tissues, was purchased from Clontech (7775-1; Lot #8110216, Palo Alto, CA, USA). A radiolabeled probe was prepared from the 500-bp *Hind*III fragment of the human clone SKhNrf3/1-1 [4]. Hybridization was performed according to the manufacturer’s instructions.

### 4.3. Cell Culture

HCT116, DLD-1, A172, and HeLa cells were cultured in DMEM (Wako Pure Chemicals, Osaka, Japan) supplemented with 10% FBS (Sigma-Aldrich, St. Louis, MO, USA) and 1% penicillin/streptomycin (Wako Pure Chemicals, Osaka, Japan). H1299 cells were cultured in RPMI1640 (Nacalai Tesque, Kyoto, Japan) supplemented with 10% FBS and 1% penicillin/streptomycin. L-WRN cells (CRL-3276; ATCC) were cultured in DMEM (Wako Pure Chemicals, Osaka, Japan) supplemented with 10% FBS. All cell lines were cultured in a humidified incubator at 5% CO_2_ and 37 °C.

### 4.4. Apc ^flox/flox^ and ApcΔ716 Mice

*Apc ^flox/flox^* (*Apc ^580S flox/580S flox^*) mice were kind gifts from Ryoji Yao [35]. Generation of *Apc*Δ716 mice was described previously [31]. All animals were housed under specific pathogen-free conditions according to the regulations of the standards for human care and use of laboratory animals of Doshisha University (A18020) and the Chiba Cancer Center Research Institute (18-6, 11 April 2018) and the guidelines for the proper conduct of animal experiments of the Ministry of Education, Culture, Sports, Science and Technology of Japan (http://www.mext.go.jp/b_menu/hakusho/nc/06060904.htm).

### 4.5. Preparation of, Culture of and Apc Gene Deletion in Intestine Organoids Derived from Apc^flox/flox^ Mice

Mouse intestine-derived organoids were generated from *Apc*^flox/flox^ mice as described previously [36]. *Apc-*deficient organoids (*Apc* KO) were generated by infecting *Apc*^flox/flox^ organoids with Cre-expressing lentivirus from the LV-*Cre* pLKO.1 plasmid encoding Cre-recombinase (25997; Addgene). Organoids were seeded on Matrigel (354234; Corning; Corning, NY, USA). *Apc*^flox/flox^ organoids were cultured in Advanced DMEM/F12 (12634; Life Technologies, Carlsbad, CA, USA) supplemented with L-WRN cell conditioned medium, 1% penicillin/streptomycin (Wako Pure Chemicals, Osaka, Japan), EGF (315-09; Peprotech, Inc. Rocky Hill, NJ, USA) and Y-27632 (a Rock inhibitor) (253-00513; Wako Pure Chemicals, Osaka, Japan). L-WRN cell conditioned medium was prepared as described in [39]. *Apc* KO organoids were cultured in Advanced DMEM/F12 supplemented with 1% penicillin/streptomycin, EGF, Y-27632 and L-glutamine (Thermo Fisher Scientific Inc. Rockford, IL, USA) in a humidified incubator at 5% CO_2_ and 37 °C.

### 4.6. Transfection

The transfection of short interfering RNA (siRNA) was performed using RNAiMAX (Thermo Fisher Scientific Inc. Rockford, IL, USA) according to the manufacturer’s protocols. Sequences of siRNAs targeting human mRNAs are listed in Appendix A.

### 4.7. RNA Extraction and Quantitative Real-Time PCR (qRT-PCR)

Total RNA was prepared using ISOGENII (Nippon Gene, Tokyo, Japan). One microgram of total RNA was utilized for cDNA synthesis using random hexamer primers (Takara Bio, Ohtsu, Japan) and Moloney murine leukemia virus (M-MLV) reverse transcriptase (Thermo Fisher Scientific Inc. Rockford, IL, USA). qRT-PCR was conducted using SYBR Premix Ex Taq II (Takara Bio, Ohtsu, Japan) and a Thermal Cycler Dice Real Time System II (Takara Bio, Ohtsu, Japan). The PCR conditions were 95 °C for 30 sec, 30 cycles of 95 °C for 5 sec, 60 °C for 30 sec and 95 °C for 15 sec, 60 °C for 30 sec and 95 °C for 15 sec. All target gene expression levels were normalized to *β-actin* or *Gapdh* expression. The sequences of the primers used are listed in Appendix A.

### 4.8. Cell Fractionation and Western Blot Analysis

To prepare whole cell extracts, cells were lysed with SDS sample buffer (50 mM Tris-HCl (pH 6.8), 10% glycerol and 1% SDS). For nuclear extracts, cells were lysed in buffer A (10 mM Tris-HCl (pH 8.0), 10 mM KCl, 0.1 mM EDTA, 1.5 mM MgCl_2_), 1x protease inhibitor (Nacalai Tesque, Kyoto, Japan), 10 µM MG132 (Peptide Institute, Osaka, Japan) and 10% NP-40 (final concentration 2%). After centrifugation, the precipitated nuclei were lysed with SDS sample buffer. The protein concentrations of the cell extracts were measured using a bicinchoninic acid (BCA) kit (Thermo Fisher Scientific Inc. Rockford, IL, USA). Proteins were separated by sodium dodecyl sulfate-polyacrylamide gel electrophoresis (SDS-PAGE) and transferred to PVDF membranes (Immobilon-P transfer membrane, Millipore, Billerica, MA, USA). The blots were incubated with the primary antibodies indicated on the figures and then with a horseradish peroxidase-conjugated secondary antibody (Thermo Fisher Scientific Inc. Rockford, IL, USA). The protein bands were visualized using enhanced chemiluminescence (GE Healthcare, Pittsburgh, PA, USA).

### 4.9. Chromatin Immunoprecipitation (ChIP)

HCT116 cells grown on a 10-cm dish were cross-linked in 1% formaldehyde for 10 min, followed by quenching with 0.125 M glycine for 5 min at room temperature and two washes with ice-cold phosphate-buffered saline (PBS). ChIP analysis was conducted as described previously [40], with minor modifications as follows: immunoprecipitated DNA was purified by phenol-chloroform extraction and ethanol precipitation, dissolved in TE and analyzed by quantitative real-time PCR. The utilized antibodies were anti-NRF3 (#9408), anti-β-catenin (610153; BD Biosciences, San Jose, CA, USA) and anti-TCF4 (C48H11; Cell Signaling Technology, Danvers, MA, USA). Melting curves of qRT-PCR show the specificity of the reaction (Appendix A). The sequences of the primers used are listed in Appendix A.

### 4.10. Cell Proliferation Assay

HCT116 cells were plated onto 6-well dishes (1 × 10^5^ cells per well), transfected with the indicated siRNA and cultured for the indicated number of days. The cells were detached from plates with 0.05% trypsin and gently suspended in ice-cold PBS. Cell counting was performed using a hemocytometer.

### 4.11. Statistical Analysis

The statistical significance of repeated measurements was evaluated using ANOVA-Tukey’s t-test and Student’s *t*-test. All values are presented as the mean ± standard deviation (SD) of at least three independent experiments.

## Figures and Tables

**Figure 1 ijms-20-03344-f001:**
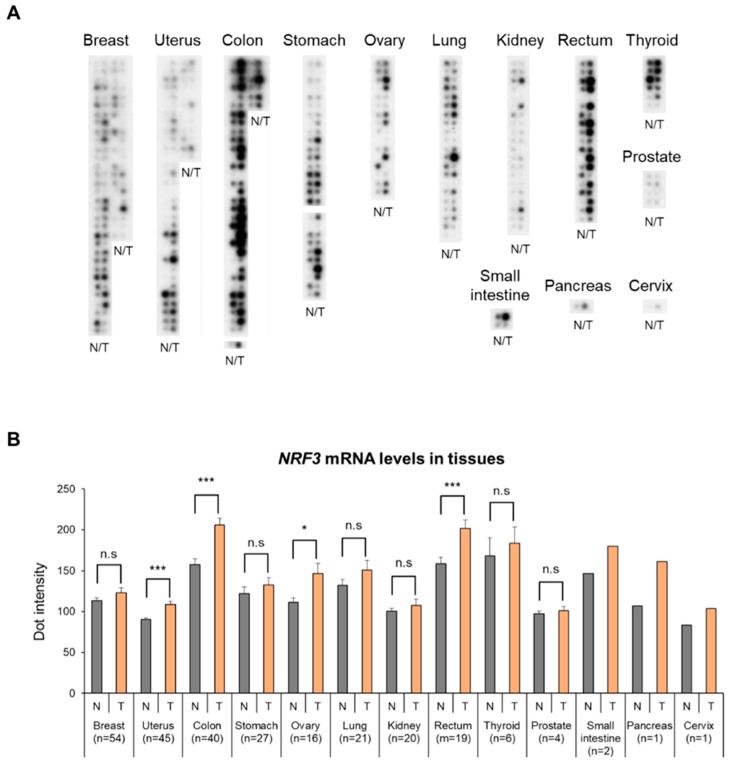
High induction of *NRF3* gene expression in human tumor tissues. (**A**) RNA dot blot analysis of *NRF3* using the Human Multiple Tissue Expression (MTE^TM^) Array. This array contains dots of polyA^+^ RNA from 76 different human tissues, normalized to ubiquitin mRNA levels. T; tumor tissue, N; adjacent normal tissue. (**B**) The bar graph shows the quantified signal intensity of each dot. The data are presented as the mean ± SD. *n*, patient number. * *p* < 0.05, *** *p* < 0.005 (*t*-test). n.s., not significant.

**Figure 2 ijms-20-03344-f002:**
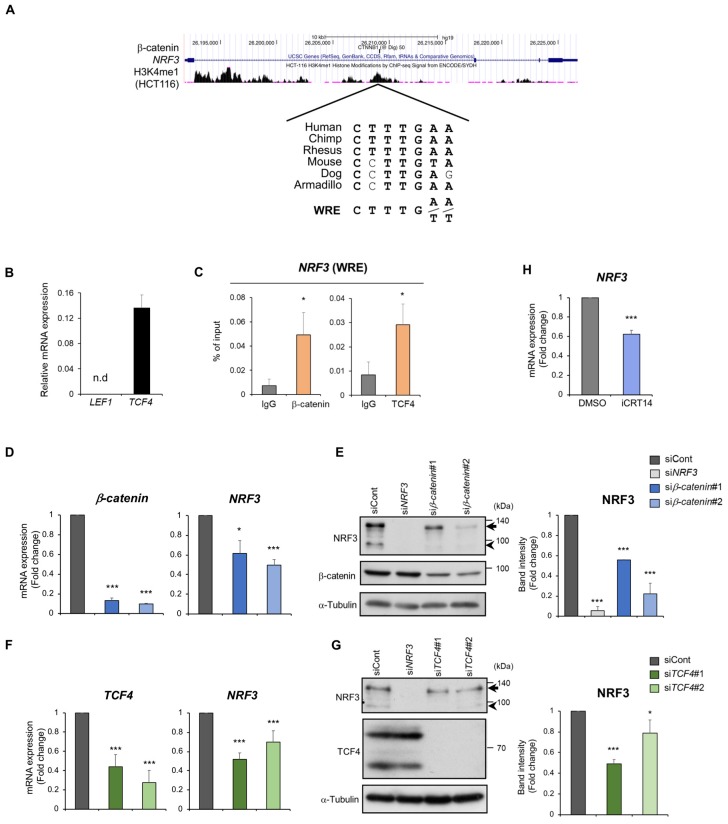
The β-catenin/TCF4 complex induces *NRF3* gene expression in colon cancer cells. (**A**) Species conservation of the WRE site (TCF/LEF consensus element) in the *NRF3* gene. ChIP peaks of β-catenin and histone H3K4me1 (a marker of a transcriptionally active locus) are shown from the UCSC Genome Browser data from ChIP-seq analyses of digestive tract-derived cells. (**B**) qRT-PCR shows the dominant expression of *TCF4* rather than *LEF1* in HCT116 cells. n.d., not detectable. (**C**) ChIP analysis using HCT116 cells indicated the binding of β-catenin and TCF4 to the WRE sequence in the *NRF3* gene. (**D**–**G**) Knockdown of *β-catenin* (**D**, **E**) or *TCF4* (**F**, **G**) significantly reduced *NRF3* expression at both the mRNA (**D**, **F**) and protein levels (**E**, **G**). Two independent siRNAs against the indicated genes were transfected into HCT116 cells, and qRT-PCR and western blot analyses were performed 48 h after transfection. The arrows and arrowheads in the immunoblots indicate the cytoplasmic and nuclear forms of NRF3, respectively. An immunoblot using an anti-TCF4 antibody showed two splicing variants of the TCF4 protein (**G**). (**H**) The β-catenin/TCF4 complex inhibitor iCRT14 also suppressed *NRF3* mRNA expression in HCT116 cells. mRNA and protein expression were normalized to *β-actin* mRNA and α-Tubulin protein expression, respectively. The data are presented as the mean ± SD (*n* = 3). * *p* < 0.05, *** *p* < 0.005 (*t*-test in C and H, ANOVA/Tukey in **D**–**G**).

**Figure 3 ijms-20-03344-f003:**
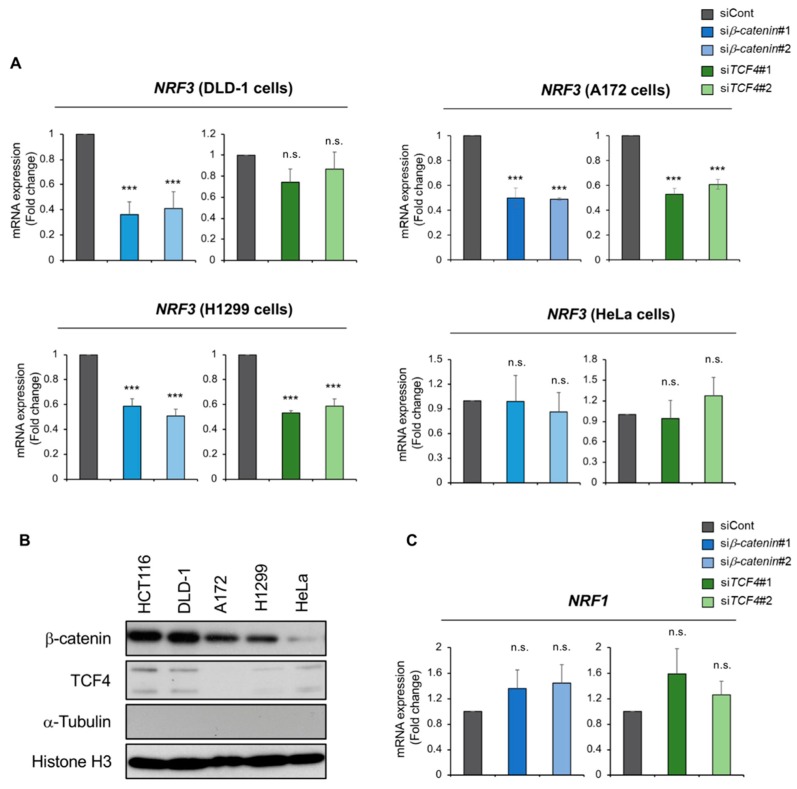
The β-catenin/TCF4 complex induces *NRF3* gene expression in cancer cell lines with mutations in the Wnt-β-catenin pathway. (**A**) Knockdown of β-catenin or TCF4 was performed in DLD-1, A172, H1299 (mutated Wnt-β-catenin system) and HeLa cells (intact Wnt-β-catenin system). The experiment was performed as described in the legend for Figure 2. (**B**) Nuclear localization of β-catenin and TCF4 in cancer cells. Nuclear extracts from the indicated cells were prepared and subjected to western blot analyses. Histone H3 served as a loading control. The absence of α-Tubulin in the extracts indicates no cytoplasmic fraction contamination. (**C**) qRT-PCR analysis of *NRF1* upon β-catenin or *TCF4* knockdown in HCT116 cells. mRNA expression levels were normalized to *β-actin* expression, and the data are presented as the mean ± SD (*n* = 3). *** *p* < 0.005, n.s., not significant (ANOVA-Tukey in (**A**,**C**)).

**Figure 4 ijms-20-03344-f004:**
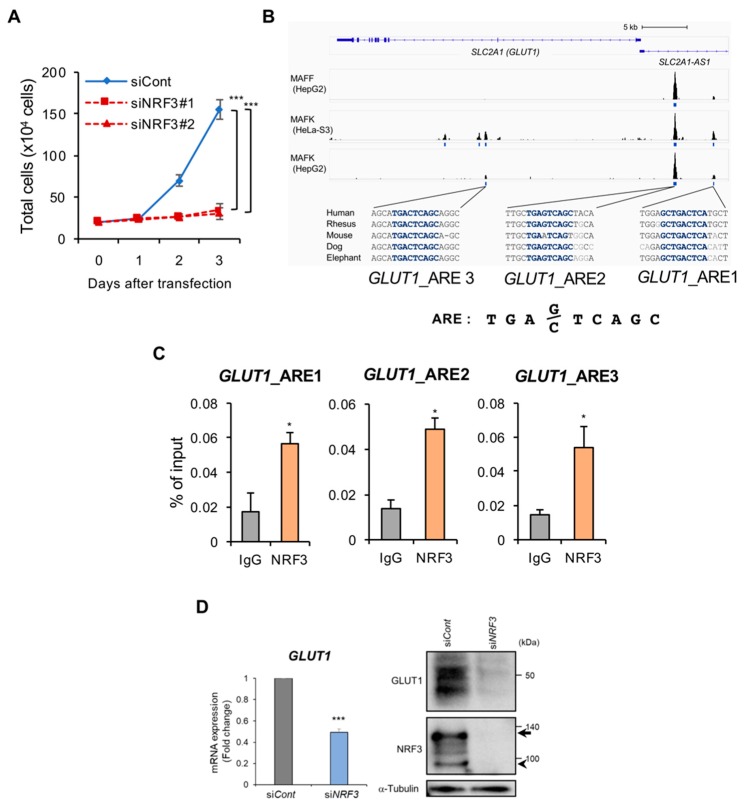
NRF3 activates cell proliferation and *GLUT1* gene expression in cancer cells. (**A**) *NRF3* knockdown significantly reduced the proliferation of HCT116 cells. The cells were transfected with control siRNA or NRF3 siRNA#1 or #2. At 1-3 days after transfection, the cell numbers were counted using a hemocytometer. The initial cell numbers at the time of transfection were 1 × 10^5^. The data are presented as the mean ± SD (*n* = 3). *** *p* < 0.005 (ANOVA/Tukey). (**B**) Three species-conserved ARE sites (NRF3 recognition sequence) were present in the promoter region and the second intron of the *GLUT1* (*SLC2A1*) gene. ChIP-seq data (SRX150386, SRX150370, SRX150689) in the ChIP-Atlas database show the binding of MAFF and MAFK as heterodimer partners of NRF3 to the AREs in the *GLUT1* gene. The blue bars represent the peak regions. The ARE sequences in the human gene are highly conserved among several species (blue). (**C**) ChIP analysis using HCT116 cells revealed NRF3 binding to three ARE sequences in the *GLUT1* gene. * *p* < 0.05 (*t*-test). (**D**) *NRF3* knockdown attenuated *GLUT1* gene expression at both the mRNA and protein levels. *NRF3* siRNA was transfected into HCT116 cells, and *GLUT1* gene expression was determined by qRT-PCR at 48 h after transfection. The arrows and arrowheads in the immunoblot indicate the cytoplasmic and nuclear forms of NRF3, respectively. RNA expression levels were normalized to *β-actin* expression, and data are presented as the mean ± SD (*n* = 3). *** *p* < 0.005 (*t*-test).

**Figure 5 ijms-20-03344-f005:**
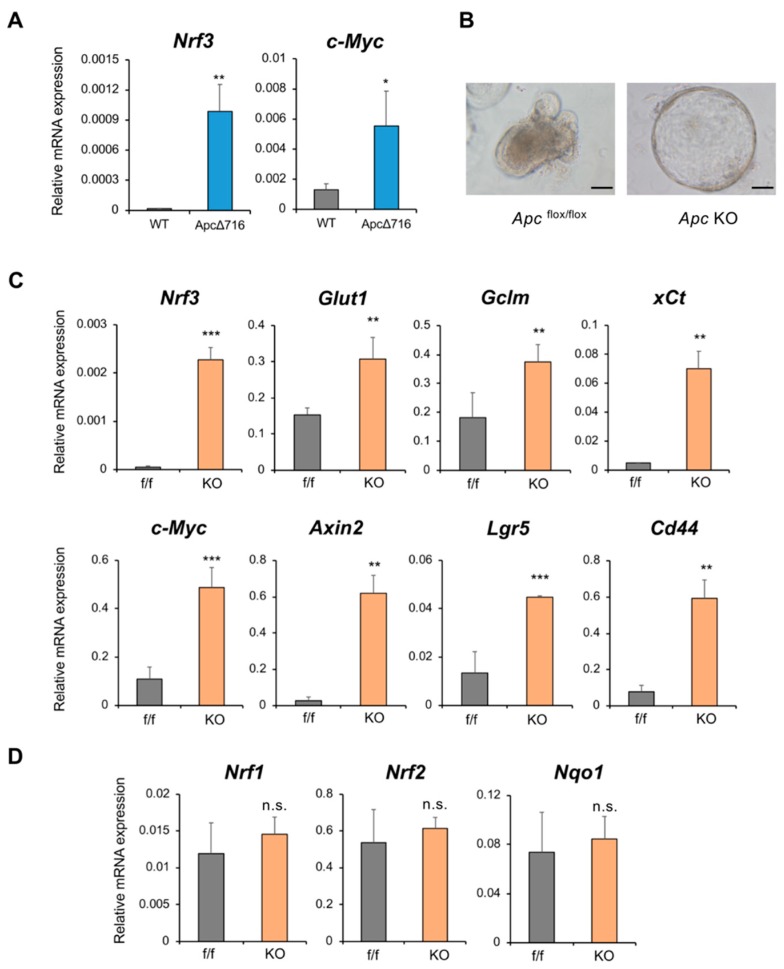
*Apc* gene deletion induces *Nrf3* gene expression in mouse intestine and organoids. (**A**) *Nrf3* expression, along with that of the β-catenin target gene *c-Myc*, was remarkably higher in intestinal tumors from *Apc*Δ716 mice than in those from wild-type mice (*n* = 4). (**B**) Appearance of intestinal epithelial organoids derived from *Apc*^flox/flox^ and *Apc* KO mice. Scale bar, 20 μm. (**C**) Loss of the *Apc* gene in organoids augmented the expression of the *Nrf3* and *Glut1* genes, along with that of β-catenin target genes (*c-Myc*, *Axin2*, *Lgr5*), as shown by qRT-PCR analysis. f/f and KO stand for *Apc*^flox/flox^ and *Apc* KO, respectively. The *Lgr5* and *Cd44* genes are stem/progenitor cell markers of intestinal epithelial cells. *Gclm* and *xCt* are ARE-containing target genes of CNC family proteins. (**D**) Gene expression of the *Nrf3*-related factors *Nrf1* and *Nrf2* was not altered*. Nqo1* is a target gene of *Nrf2.* The mRNA expression of each gene was normalized to *Gapdh* expression, and the data are presented as the mean ± SD (*n* = 3). * *p* < 0.05, ** *p* < 0.01, *** *p* < 0.005, n.s., not significant (*t*-test).

**Figure 6 ijms-20-03344-f006:**
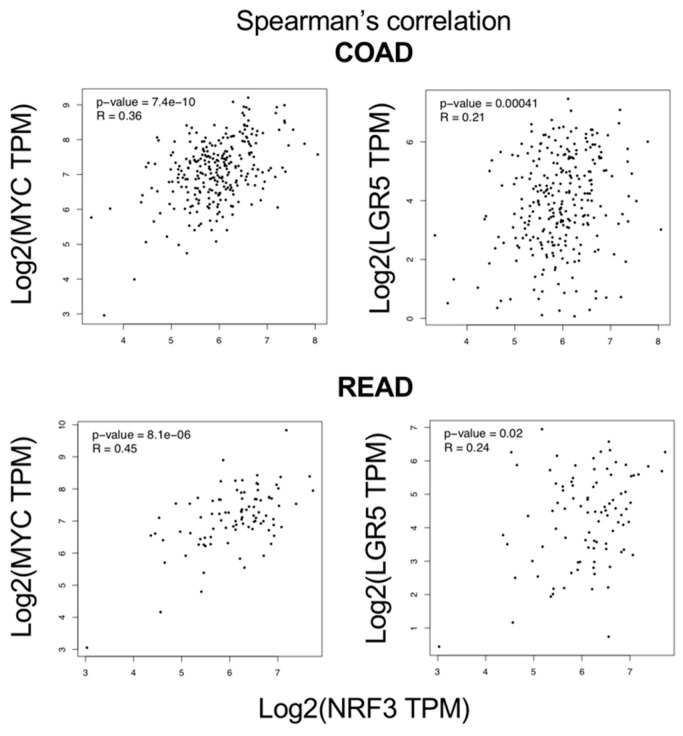
Correlation of *NRF3* gene expression and that of the β-catenin target genes *c-MYC* and *LGR5* in human colorectal cancer. Spearman’s correlation plots represent the colorectal adenocarcinoma (COAD) and rectal adenocarcinoma (READ) results from the GEPIA database.

**Figure 7 ijms-20-03344-f007:**
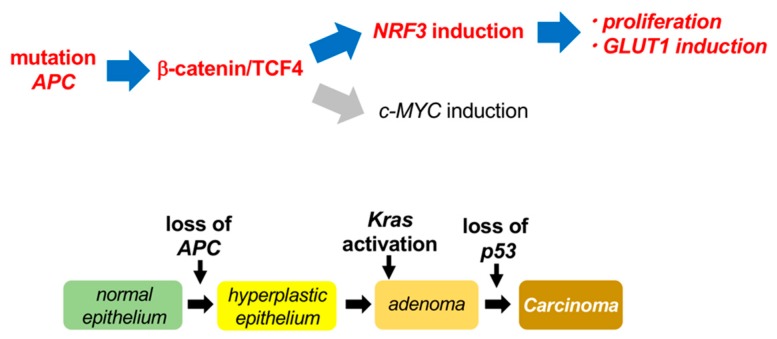
Schematic model of the β-catenin/TCF4-NRF3 axis in cancer cells. *APC* gene deletion is a crucial first step in the initiation of human sporadic colorectal cancer [17]. This deletion activates β-catenin protein and, consequently, the TCF4-mediated expression of tumor-related genes such as the protooncogene *c-MYC*. This study showed that β-catenin/TCF4 also activates *NRF3* mRNA expression, which leads to the upregulation of proliferation and *GLUT1* gene expression in cancer cells. These results define a new biological function of the β-catenin/TCF4-NRF3 axis in cancer cells.

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
