# Peer review of "β-Catenin/TCF4 Complex-Mediated Induction of the NRF3 (NFE2L3) Gene in Cancer Cells"

_ijms, 2019, doi:10.3390/ijms20133344_

Reviewer 1 Report

The authors suggested NRF3 should be upregulated by WNT/beta-catenin pathway and NRF3 might contribute to the proliferation of cancer cells possibly via modulation of cancer cell metabolism. To support this, the authors employed and performed appropriate methods and so, this manuscript is overall well designed and written.

However, I should raise some issues to be published in IJMS.

First of all, almost all data are shown in the level of mRNA, which is short to lead to the conclusion. So, the qPCR data must be shown with western blot analysis.

Secondly, since as the authors stated in this manuscript, NRF3 is usually degraded in the basal state. the upregulation of NRF3 by WNT stimulation should be demonstrated at the level of protein.

Thirdly, to stimulate the expression of target genes of NRF3 in WNT/beta-catenin pathway, NRF3 should be translocated to nuclei in response to WNT3a treatment. This phenomenon should be also contained in the results.

Minor points

Figure 1B. Comparison between LEF1 and TCF4 expression at the level of protein and their localization in the nuclei should be needed.

Figure 1G. Which band is the TCF4 protein?

Figure 4C. qPCR data of ChIP assay should be shown along with conventional PCR images or melting curve of each qPCR reaction should be supplemented at least.

Figure 4D. The down-regulation of GLUT1 by siRNA against NRF3 should be rescued by overexpression of NRF3 to confirm that GLUT1 is a transcriptional target of NRF3.

In Materials and Methods, the method of qPCR should be described in detail.

Reviewer 2 Report

This work revealed that NRF3 induction in cancer cells is under control of the b-catenin pathway.  While this is a quite interesting study, there remain several critical apprehensions regarding the adequacy of the conclusion.  Following points need to be considered upon revision of this manuscript.

Major points

1.     In figure 2, reduction of NRF3 mRNA by b-catenin knockdown and binding of NRF3 to the WRE sequence in the NRF3 gene locus were shown. However, necessity of the WRE sequence in the NRF3 for NRF3 expression was not examined. The authors should delete the WRE sequence by genome editing method and examine the effects on NRF3 expression.

2.     Does the WRE sequence in the NRF3 gene locate in intron of the gene? The location should be mentioned in text. In addition, ChIP peaks of histone H3K4me1 from the UCSC Genome Browser data are shown, but there is no mention about this information in text. The explanation should be added in text.

3.     Only mRNA level of NRF3 was shown in the experiment with iCRT14 in figure 2H. Protein level of NRF3 also should be examined in this experiment.

4.     In figure 3, knockdown of TCF4 did not reduce NRF3 expression in DLD-1 cells, implying low TCF4 expression in DLD-1 cells. As is the case of b-catenin in figure 3B, TCF4 protein levels should be compared among the five cell lines.

5.     In figure 3, NRF2 expression was reduced by knockdown of b-catenin and TCF4 in HCT116. Is figure 3C necessary? This result is not consistent with the observation that NRF2 expression was not affected by deletion of Apc gene in figure 5D. If the authors consider that NRF2 also can be regulated by b-catenin and TCF4, effects of knockdown of b-catenin and TCF4 on NRF2 expression using DLD-1 cells, A172 cells, H1299 cells and HeLa cells should be examined to confirm the consistency.

6.     Since p53 is reported to down-regulate GLUT1 (Schwartzenberg-Bar-Yoseph et al, 2004), there is the possibility that NRF3 suppresses GLUT1 expression via down-regulation of p53. To exclude the possibility, the authors should show that reduction of GLUT1 by NRF3 knockdown cannot be cancelled by simultaneous knockdown of p53 (figure 4D).

7.     In figure 5C, elevated expression levels of Gclm and xCT in Apc KO intestine were shown. Does it mean that Gclm and xCT are NRF3 target genes? Since Gclm and xCT are well-known target genes for NRF1 and NRF2, another possibility is that NRF1 and/or NRF2 could be activated in Apc KO intestine although typical NRF2 target gene NQO1 was not induced in Apc KO intestine (figure 5D). It would be nice if expression of UHMK1, which was previously identified as Nrf3 target gene by the authors (Chowdhury et al, 2017), is examined in the Apc KO intestine.

Author Response

Round  2

Reviewer 1 Report

The authors cleared the issues raised by the reviewers.

Just two corrections as follows seem to be needed for the publication of this new version.

In line 188, Supplementary Figure 2 must be Supplementary Figure 3.

Supplementary Figure 2 had better be mentioned in Materials and Methods 4.7.

Reviewer 2 Report

The reviewer is satisfied with the authors' response.
